# Non-Destructive Detection of Strawberry Quality Using Multi-Features of Hyperspectral Imaging and Multivariate Methods

**DOI:** 10.3390/s20113074

**Published:** 2020-05-29

**Authors:** Shizhuang Weng, Shuan Yu, Binqing Guo, Peipei Tang, Dong Liang

**Affiliations:** National Engineering Research Center for Agro-Ecological Big Data Analysis & Application, Anhui University, 111 Jiulong Road, Hefei 230601, China; yushuan_1994@163.com (S.Y.); guobingqing65@gmail.com (B.G.); gdsama1101@163.com (P.T.); comm_2006@foxmail.com (D.L.)

**Keywords:** hyperspectral imaging, multiple features, strawberries, SSC, pH, VC, multivariate methods

## Abstract

Soluble solid content (SSC), pH, and vitamin C (VC) are considered as key parameters for strawberry quality. Spectral, color, and textural features from hyperspectral reflectance imaging of 400–1000 nm was to develop the non-destructive detection approaches for SSC, pH, and VC of strawberries by integrating various multivariate methods as partial least-squares regression (PLSR), support vector regression, and locally weighted regression (LWR). SSC, pH, and VC of 120 strawberries were statistically analyzed to facilitate the partitioning of data sets, which helped optimize the model. PLSR, with spectral and color features, obtained the optimal prediction of SSC with determination coefficient of prediction (R_p_^2^) of 0.9370 and the root mean square error of prediction (RMSEP) of 0.1145. Through spectral features, the best prediction for pH was obtained by LWR with *R_p_*^2^ = 0.8493 and RMSEP = 0.0501. Combination of spectral and textural features with PLSR provided the best results of VC with *R_p_*^2^ = 0.8769 and RMSEP = 0.0279. Competitive adaptive reweighted sampling and uninformative variable elimination (UVE) were used to select important variables from the above features. Based on the important variables, the accuracy of SSC, pH, and VC prediction both gain the promotion. Finally, the distribution maps of SSC, pH, and VC over time were generated, and the change trend of three quality parameters was observed. Thus, the proposed method can nondestructively and accurately determine SSC, pH, and VC of strawberries and is expected to design and construct the simple sensors for the above quality parameters of strawberries.

## 1. Introduction

Strawberry, which is widely known as the queen of fruit, is popular worldwide for its good taste and rich nutrients [1]. Strawberry is not only rich in sugar, acid, and cellulose but also contains vitamins, such as vitamin C (VC), which helps promote the synthesis of fats and proteins and maintain the immune function of humans [2]. Soluble solid content (SSC) and pH are the key parameters for assessing strawberry taste, maturity, and harvest time [3], and VC is the monitored nutritional indicator of strawberry. SSC, pH, and VC are often used to classify strawberry grade [4]. The common detection methods for SSC, pH, and VC are based on refractive index, hydrogen ion concentration, and high-performance liquid chromatography [5,6,7]. However, these methods are time-consuming, laborious, destructive, and cannot be used for massive detection. Hence, developing a simple and non-destructive method for detecting SSC, pH, and VC is significant.

Non-destructive and widely used analytical methods in the field of food quality detection mainly include machine vision and spectroscopy. Machine vision can stably obtain external features, such as color and texture, in a non-contact manner. It has been used to classify the fat of meat and monitor shrimp color [8]. However, machine vision cannot detect the internal quality of the target. Among spectroscopic methods, visible/near-infrared (Vis/NIR) spectroscopy is usually used for the rapid analysis of food quality and unconstrained by the target morphology. Some physical and chemical parameters, such as wheat and corn hardness [9], black bean color and texture [10], total antioxidant of gluten-free grains [11], and the protein, starch, oil, and total polyphenol of faba bean seeds [12], have been detected using Vis/NIR spectroscopy. However, spectral methods only provide the “spot” information of objects which limits acquisition of local or global characteristics.

Hyperspectral imaging (HSI) combines the advantages of machine vision and spectroscopy to provide spectral and spatial information about the external and internal qualities of food, which has been applied in defects of peaches [13], the color of sausage [14], the SSC and acidity of limes [15], and the VC of jujube [7]. In addition, many researchers have used HSI to study the qualities of strawberries, including its moisture content, pH, SSC, and firmness [4,16,17], bruise and fungi contamination [18,19], ripeness [3,20], foliar anthracnose [21], and decay [1]. In these studies, however, only the spectra extracted from the HSI images were used for the above analysis. Among them, Ding et al. [17] detected the SSC of strawberries with determination coefficient of prediction (*R_p_*^2^) of 0.8519, and ElMasry et al. [4] got the best result of *R_p_*^2^ = 0.8836 for pH of strawberries. In fact, other features, such as color, texture, and morphology, may also be easily extracted from HSI images. The combination of various features helps improve the accuracy of analysis, such as classifying pork muscles [8], predicting the total volatile basic nitrogen of chicken [22], and determining the S-ovalbumin content in stored eggs [23]. Considering the characteristics of strawberries, the spectral, color, and textural features were combined for the analysis of SSC, pH, and VC. To the best of our knowledge, no such study on strawberries has been performed.

This study aims to explore the feasibility of the non-destructive detection of SSC, pH, and VC in strawberries through integration of the spectral, color, and textural features extracted from hyperspectral images and multivariate methods. First, the average spectra were extracted from the hyperspectral images of strawberries, and wavelet transform (WT) and multivariate scatter correction (MSC) were used to remove the spectral noise. Subsequently, the color and textural features were obtained using the color moment, the gray level-gradient co-occurrence matrix (GLGCM), and the Gabor filter. Partial least-squares regression (PLSR), support vector regression (SVR), and locally weighted regression (LWR) were adopted to develop the regression models for SSC, pH, and VC prediction. Competitive adaptive reweighted sampling (CARS) and uninformative variable elimination (UVE) were used to select the important variables of spectral, color, and textural features to obtain a robust analysis. Finally, the distribution maps of SSC, pH, and VC over storage time were generated based on the pixel-wise spectra and the established model.

## 2. Materials and Methods

### 2.1. Samples Preparation

A total of 120 strawberries (“*Hongyan*”) of full-red and defect-free were picked from a local fruit planting base in Changfeng of Anhui Province (China, 32°10′ N, 117°20′ E) in March 2019. All strawberries were individually rinsed, wiped, numbered, and stored at 17–19 °C before the experiment. Hyperspectral images of strawberries were acquired in a laboratory next to the fruit planting base. 

### 2.2. Acquisition of Hyperspectral Images

A hyperspectral reflectance imaging system (Appendix A) was used to acquire hyperspectral images of the strawberries. This system consisted of a high spectrograph (SOC710VP, SOC, San Diego, CA, USA), a charge coupled device (CCD) camera, two tungsten halogen lamps, a moving platform, a black box, and a computer with image acquisition software [24]. The camera resolution was 692 × 692, and the resolution of the spectra was 2.31 nm within the spectral range of 374–1020 nm. The wavelength range of the light source was 300–2500 nm. The exposure time of the CCD camera was 0.06 s. The moving platform was adjusted so that the distances from the samples to the lamps and lens were 395 and 350 mm, respectively. Strawberries at 17–19 °C generally go bad after 3 days [25]. To increase the sample variability, the HSI images of 24 strawberries were collected every 10 h within 40 h, and the images of 120 strawberries in five storage time were obtained. After collection of HSI images, and the reference values of SSC, pH, and VC were measured.

Owing to the uneven light distribution and dark current from the camera, the hyperspectral images required calibration prior to spectral extraction. The raw images were corrected using the following Equation (1):(1)$$R=\frac{R_{raw}-R_{d}}{R_{w}-R_{d}}$$
where *R* is the corrected image, *R_raw_* is the raw hyperspectral image, *R_w_* is the white reference image acquired from a white board with 99% reflectance, and *R_d_* is the dark reference image obtained through the shutting lamps.

### 2.3. Reference Values Measurement of SSC, pH, and VC

The reference values of SSC, pH, and VC of the strawberries were determined by traditional destructive methods. First, the strawberry was placed in a juicer and completely squeezed. Then, the juice was poured into test tube for measurement. A total of 120 strawberry juices were prepared for measuring their SSC, pH, and VC. SSC and pH were measured by a digital refractometer (CHT65, Lohand Biological Co. Ltd., HangZhou, China) and a pH meter (206pH1, Testo SE & Co. KGaA, Baden-Württemberg, Germany) [26], respectively. An ultraviolet-visible spectrophotometer (U-2910, Hitachi, Japan) was used to obtain the absorbance value of VC at 250 nm, and the VC concentration was calculated with the calibration curve (Appendix A) based on Lambert-Beer law [27,28,29], and then the VC content was calculated according to the following Equation (2):(2)$${\mathrm{VC}\;\mathrm{content}\;(g\;\mathrm{kg}}^{-1})=\frac{C\times V}{M}\times 1000$$
where *C* is the VC concentration, *V* is the volume of the juice and *M* is the weight of strawberry.

### 2.4. Extraction of the Spectrum, Color, and Texture of Strawberries

#### 2.4.1. Extraction of Spectral Features

After image correction, binary images were extracted using an image thresholding segmentation method based on the super-red color characteristic (Figure 1). The region of interest (ROI) images of the strawberries were then obtained by applying the binary images to the corrected hyperspectral images. The mean spectrum was calculated by averaging the spectra of all pixels in the ROI for further analysis.

The entire spectrum obtained was resized into a useful spectral region of 400–1000 nm with 260 bands because the spectra in the wavelength ranges of 374–400 nm and 1000–1020 nm contained large amounts of noise [26,30]. 

#### 2.4.2. Extraction of Color and Textural Features

Color and textural features were extracted from the ROI images. The color features could be extracted using color histogram and moment. While color histograms cannot differentiate local color information in the image, any color distribution in the image can be represented by its moment. Given that the color distribution information is mainly concentrated in the low-order moment, the first-, second-, and third-order moments are sufficient to express the color distribution of the image. Color moment was adopted in the RGB color space, and images of the red (691 nm), green (531 nm), and blue (457 nm) bands were selected from the hyperspectral images to obtain color features (Figure 1). Overall, nine color features were extracted from a single strawberry.

The extraction of textural features was divided into two steps. Initially, the weight of each band of 260 bands was obtained according to CARS. The wavelengths with a large weight contained more information, and the first two wavelengths which are of largest weight correlated with SSC, pH, and VC are selected. The selected wavelengths were 676 and 910 nm for SSC, 444 and 842 nm for pH, and 837 and 891 nm for VC (Appendix A). Then, the GLGCM [31] and the Gabor filter [32] were used to extract the textural features of the images of the two corresponding selected bands. Fifteen features of GLGCM contained comprehensive grayscale and gradient information. The Gabor filter extracted three features at the scale with the horizontal axis of 2 and vertical axis of 4, and the direction of 60° in the frequency domain. For a single strawberry, 36 textural features were obtained (Figure 1).

### 2.5. Spectra Pretreatment and Variable Selection Methods

Given the background and noise distraction of raw spectra, WT, and MSC were used to remove irrelevant spectral information and stabilize the models. WT provides a time–frequency window that has an excellent performance in signal processing [33]. Moreover, MSC can effectively eliminate the spectral difference caused by the scattering effect of light.

Selection of important variables is important to reduce model complexity and experiment costs. CARS selects variables with high absolute values of weight in the model through adaptive reweighted sampling. The method obtains the variable subset of the lowest root mean square error of cross-validation (RMSECV) with interactive verification, this subset combines optimal variables. CARS has been applied to many fields, such as selecting variables related to the SSC of apples and pears [30,34], the pH of honey [35], and the VC of jujube [7]. UVE is a variable selection method based on the analysis of PLSR coefficients. The approach can evaluate the reliability of each variable in the model through a variable selection criterion, and eliminate uninformed variables.

### 2.6. Models Establishment and Performance Evaluation

The models for predicting the SSC, pH and VC were developed by PLSR, SVR, and LWR. PLSR projects the predicted and observed variables into a new space to find a linear regression model. PLSR essentially decomposes the independent (**X**) and the dependent (**Y**) variable data matrices according to the covariance maximization criterion and establishes a regression relationship equation between the interpretive implicit and reaction hidden variables. PLSR is especially suitable for **X** with more variables than **Y** and for considering multicollinearity in the variables of **X** [36]. SVR is based on Vapnik–Chervonenkis dimensional of statistical learning theory and the minimum principle of structural risk. It seeks the best compromise between the complexity of a model and the learning ability based on the sample information to obtain the best generalizability of the model. SVR realizes linear regression by constructing a linear decision function in high-dimensional space after dimension elevation [37]. It is useful in solving the regression problem of high-dimensional features and works well when the feature dimension is larger than the sample size. LWR is a nonparametric method that retrains adjacent data to obtain new parameter values each time a new sample is predicted. Ordinary linear regression strives to find a model that minimizes the global cost function, which causes local data under-fitting. However, the cost function designed by LWR increases the weight of the points near the predicted point, the weight decreases as the distance increases, and LWR effectively avoids the under-fitting problem [38]. 

The spectra of 24 strawberries in every storage time were split into two groups with the ratio of 3:1 according to the Kennard–Stone algorithm [39]. In total, spectra of 90 strawberries were selected as the calibration set to develop the models, and the spectra of remaining 30 strawberries were used as the prediction set to estimate the performance of the models. The SSC, pH, and VC statistics of the 120 strawberries are depicted in Table 1, and the change ranges of the calibration sets were higher than those of the prediction sets, which is consistent with statistical principles. The determination coefficients of calibration (*R_c_*^2^) and prediction (*R_p_*^2^) and the root mean square error of calibration (RMSEC) and prediction (RMSEP) were employed to evaluate the performance of the models. The above processing was conducted in MATLAB (Version: R2014a, Mathworks, Inc., Natick, MA, USA).

## 3. Results and Discussion

### 3.1. Spectral Characteristics

The reflectance spectra of the strawberries over the wavelength range of 400–1000 nm were obtained from the hyperspectral images and are shown in Figure 2A. The bands reflect information on the physical parameters and chemical and biological characteristics of the strawberries. The band at around 680 nm can be related to chlorophyll [4], while the band at around 750 nm is ascribed to the third overtone of water [40]. In the NIR region, the band near 960 nm is due to the O–H of water [41]. The reflectance spectra of strawberries with different SSCs, pH, and VC contents are shown in Figure 2B,D. The figures show that the overall spectral reflectance increases with increasing SSC in range of 750–950 nm; however, the spectral change trend of pH and VC shows the opposite trend. Moreover, with regard to SSC and pH, the spectral change at 400–750 nm is inconsistent with that at 750–950 nm, presenting a partial overlap. The spectral reflectance at 400–750 nm and the VC content are still inversely proportional. In other words, the spectra of strawberries differ according to their SSCs, pH, and VC contents. This phenomenon preliminarily proves the feasibility of detecting SSC, pH, and VC in strawberries using the reflectance spectra extracted from HSI.

### 3.2. Detection of SSC, pH, and VC Using Spectroscopy

The spectra of 260 bands were extracted from the hyperspectral images and processed by WT and MSC to reduce noise. The spectra pretreated by WT (Appendix A) were smoother than the raw spectra. Then, PLSR, SVR, and LWR were used to develop the regression models to predict the SSC, pH, and VC of the strawberries. The analysis results are shown in Table 2 and detailed parameter settings are shown in Appendix A. 

For the prediction of SSC, PLSR with raw spectra revealed the prediction results: *R_c_*^2^ = 0.9769, RMSEC = 0.0657, *R_p_*^2^ = 0.9044, and RMSEP = 0.1411. WT improved the performance of the PLSR models and produced results of *R_c_*^2^ = 0.9776, RMSEC = 0.0647, *R_p_*^2^ = 0.9182, and RMSEP = 0.1305. SVR performed poorly with the raw and WT-processed spectra. LWR and WT provided the best prediction ability for SSC with *R_c_*^2^ = 0.9999, RMSEC = 0.0007, *R_p_*^2^ = 0.9364, and RMSEP = 0.1151. During pH analysis, WT consistently deteriorated the performance of the regression models. LWR with the raw spectra showed the best prediction ability with *R_c_*^2^ = 0.9819, RMSEC = 0.0181, *R_p_*^2^ = 0.8493, and RMSEP = 0.0501. The model developed with SVR and MSC exhibited better performance for predicting VC with *R_c_*^2^ = 0.9243, RMSEC = 0.0180, *R_p_*^2^ = 0.8415, and RMSEP = 0.0247. Thus, multivariate methods and spectra from the hyperspectral images could be used to detect SSC, pH, and VC in strawberries. Besides their spectra, the color, texture, and morphology of analytes could be easily extracted from the hyperspectral images [8,22,23,42], and they are often combined with spectroscopy for the analysis of properties. Compared to single feature, the integration of multiple features can generally improve the detection accuracy [22,23]. Considering the specific characteristics of SSC, pH, and VC in strawberries, the color and textural features were further extracted and integrated in the follow-up analysis.

### 3.3. Color and Textural Features of Strawberries

First, the nine color features of strawberries were extracted from the hyperspectral images by the first, second, and third color moments, which reflect the brightness, darkness, distribution range, and distribution symmetry of the image color. Textural features from the images of the first two largest weight wavelengths (Figure 1) selected by CARS (detailed parameter settings are shown in Appendix A) were then extracted through the Gabor filter and GLGCM, and the selected wavelengths have the higher correlation with SSC, pH, and VC. The textural features include mean, contrast and entropy, small grads dominance, big grads dominance, gray asymmetry, grads asymmetry, energy, gray mean, grads mean, gray variance, grads variance, correlation, gray entropy, grads entropy, entropy, inertia, and homogeneity; here, grads dominance can reflect the degree of grayscale change of the image. A total of 36 textural features were obtained for each strawberry. 

The normalized color features and textural characteristics of 676 nm of strawberries with different SSCs are shown in Figure 3. Figure 3A illustrates that the values of color features first increase and then decrease with increasing SSC. The uniform change trend observed indicates that the information of the nine color features has some redundancy. By contrast, changes in different textural features are inconsistent (Figure 3B). Strawberries with different pH and VC contents also show different color and textural features (Appendix A). Therefore, color and textural features can provide supplementary, diverse, and potentially effective information for detecting SSC, pH, and VC in strawberries [8,23].

### 3.4. Detection of SSC, pH, and VC Based on Spectral, Color, and Textural Features

Spectral, color, and textural features were integrated to develop the regression models established by PLSR, SVR, and LWR for analysis of SSC, pH, and VC. Spectral data which gets the optimal results in Table 2 was adopted for the above fusion. The detailed results are shown in Table 3, and the detailed parameter settings are shown in Appendix A. During SSC analysis, addition of multiple features greatly enhances the performance of SVR models, which is still poorer than that of the PLSR and LWR models. With spectral and color features, PLSR obtains the optimal prediction of SSC with *R_c_*^2^ = 0.9800, RMSEC = 0.0611, *R_p_*^2^ = 0.9370, and RMSEP = 0.1145. However, the performance of LWR model declines with increasing features. Similarly, during determination of pH, introduction of various features do not improve the analysis results. The spectral and textural features provide the better analysis results of VC, and the optimal model is developed by PLSR (*R_c_*^2^ = 0.9567, RMSEC = 0.0138, *R_p_*^2^ = 0.8769, and RMSEP = 0.0279). The optimal prediction results for SSC, pH, and VC in the calibration and prediction sets based on the full-range spectra, color, and textural features are shown in Figure 4A–C. The figures indicate that the prediction errors of the three quality parameters are all low, and most of the data points are concentrated near the fitting line. The effects of various features on the detection of the SSC, pH, and VC of strawberries differ. Given the appropriate features and multivariate methods, an accurate analysis of the three quality parameters can be obtained.

### 3.5. Detection of SSC, pH, and VC Using Important Variables of Multiple Features

High-dimension spectral, color, and textural features inevitably contain redundant, dependent, and correlational information [22,23,43]. Such information can disturb the accuracy and robustness of analysis and increase the complexity of calculation [23,44]. Selection of important variables from the above three features helps avoid these negative effects. Thus, CARS and UVE were employed to select the important variables from the optimal feature combination for SSC, pH, and VC in the above experiments. Specifically, the variables of the higher weight in spectral, color, and textural features were sought out. The relationship between the RMSECV and the number of variables obtained with CARS is shown in Appendix A.

The analysis results are shown in Table 4, and detailed parameter settings are shown in Appendix A. For the three quality parameters, the prediction effect of CARS was better than UVE. Based on the variables selected by UVE (detailed parameter settings are shown in Appendix A), the prediction accuracy of VC improved, the prediction results of *R_c_*^2^ = 0.9591, RMSEC = 0.0134, *R_p_*^2^ = 0.9102, and RMSEP = 0.0251 were obtained from PLSR with the spectra of 31 wavelengths and 28 textural features. However, the prediction accuracy of SSC and pH deteriorated. For the important variables CARS selected, the prediction performance of SSC, pH, and VC exhibited obvious improvements. The model developed by spectra of the 59 wavelengths selected by CARS (Figure 5A) and 3 color features of the first moment presented the best analysis results of *R_c_*^2^ = 0.9445, RMSEC = 0.1076, *R_p_*^2^ = 0.9431, and RMSEP = 0.0895 for SSC by PLSR. The optimal performance for pH was obtained from the LWR models developed by spectra of the 45 wavelengths selected by CARS (Figure 5B) with *R_c_*^2^ = 0.9934, RMSEC = 0.0447, *R_p_*^2^ = 0.8858, and RMSEP = 0.0108. Moreover, compared to the methods in the previous studies [4,17], the effect of multiple features was superior. 

For VC, the lowest prediction errors are *R_c_*^2^ = 0.9228, RMSEC = 0.0185, *R_p_*^2^ = 0.9109, RMSEP = 0.0237, and the results are got from PLSR model developed by the spectra of 27 wavelengths selected by CARS (Figure 5C) and 23 textural features. The 23 features include 18 textural features from the image of 837 nm and low grads dominance, grads asymmetry, gray variance, correlation, and grads entropy from the image of 891 nm. 

As seen in Figure 5, the important wavelengths for SSC and pH are distributed in the range of 400–1000 nm, and the selected wavelengths for VC are concentrated in the range except 560–655 nm. The concrete selected wavelengths are shown in Appendix A, the number of shared important wavelengths for SSC and pH, SSC and VC, and pH and VC are 17, 4, and 5 [26,40], respectively. Moreover, 980 nm is the one shared wavelength for SSC, pH, and VC. Comparing with the results after variable selection, the gap between *R_c_*^2^ and *R_p_*^2^ is larger for the results before variable selection, thereby indicating that redundant variables of the original data may cause model over-fitting [23]. The detailed prediction results of the optimal models for SSC, pH, and VC are shown in Figure 4D–F. The figures show that the prediction errors of the optimal models of SSC, pH and VC were smaller after variable selection. Overall, the important variables selected by CARS from the feature combination and PLSR or LWR not only improve the analysis accuracy of the SSC, pH, and VC in strawberries but also simplify the models for higher robustness. The simplified models can be applied to design the multispectral systems for online detection of SSC, pH, and VC of strawberries.

### 3.6. Visualization of SSC, pH, and VC of Strawberries Over Storage Time

As for the high moisture, naked pulp, strong respiration and sensitive to fungal invasion, strawberries generally deteriorated after 3 days at 17–19 °C [25,45]. The change trend of SSC, pH, and VC of strawberries over storage time was further investigated in this study. Firstly, the HSI images of strawberries within storage time of 0 h, 10 h, 20 h, 30 h, and 40 h were measured. Then, pixel-wise spectra were extracted from the images, and the predicted values of SSC, pH, and VC were obtained based on the optimal models developed with the spectra of the selected wavelengths (Appendix A). Moreover, the predicted values were intuitively displayed with the color scale in different colors from blue (low value) to red (high value). The distribution maps for the SSC, pH, and VC of strawberries are shown in Figure 6. From figure, the SSC first rises then descends, which is the combined action of hydrolysis of carbohydrates and respiration. Hydrolysis of carbohydrates increases the SSC, but respiration consumes soluble solids continuously. Consequently, the SSC initially increases and then decreases [46]. The pH values continued to decrease with increasing storage time because of the moisture loss of strawberry [46]. The VC content also showed a declining trend, which is mainly due to the easy oxidation of VC under the action of various enzymes and air [47]. The result showed that the hyperspectral imaging can monitor the changes of strawberry quality in real time, and also provided a new technical approach for the follow-up research on shelf life of strawberry.

## 4. Conclusions

In this study, the spectral, color, and textural features extracted from hyperspectral images were investigated for the non-destructive detection of SSC, pH, and VC in strawberries. PLSR, SVR, and LWR were adopted to develop regression models for determining SSC, pH, and VC using these three types of features. WT and MSC were used to reduce the noise of the spectra. The best prediction of SSC was obtained by PLSR, and features of the WT-processed spectra and color produced *R_c_*^2^ = 0.9800, RMSEC = 0.0611, *R_p_*^2^ = 0.9370, and RMSEP = 0.1145. During pH analysis, LWR and raw spectra obtained the best results of *R_c_*^2^ = 0.9819, RMSEC = 0.0181, *R_p_*^2^ = 0.8493, and RMSEP = 0.0501. The optimal model for VC was developed by PLSR and the WT-processed spectral and textural features (i.e., *R_c_*^2^ = 0.9567, RMSEC = 0.0138, *R_p_*^2^ = 0.8769, RMSEP = 0.0279). Furthermore, CARS and UVE were employed to select important variables from the optimal feature combination for simple and robust analysis. Based on these important variables selected by CARS, the analysis accuracy of SSC, pH and VC all gained the promotion. Particularly, *R_c_*^2^ = 0.9445, RMSEC = 0.1076, *R_p_*^2^ = 0.9431, and RMSEP = 0.0895 were predicted for SSC, and *R_c_*^2^ = 0.9934, RMSEC = 0.0447, *R_p_*^2^ = 0.8858, and RMSEP *=* 0.0108 for the pH detection. VC obtained the following results: *R_c_*^2^ = 0.9228, *RMSEC =* 0.0185, *R_p_*^2^ = 0.9109, and RMSEP = 0.0237. The distribution maps of SSC, pH, and VC over storage time show the change trend of three quality parameters clearly. SSC first increased and then decreased, while pH and VC content continuously declined. Taking the results together, suited integration of multi-features in HSI and multivariate methods can achieve the non-destructive and accurate sensing systems of SSC, pH, and VC in strawberries. The proposed method can be extended to the non-destructive analysis of other quality parameters, such as firmness and fiber content of strawberries or quality of other agricultural crops. 

## Figures and Tables

**Figure 1 sensors-20-03074-f001:**
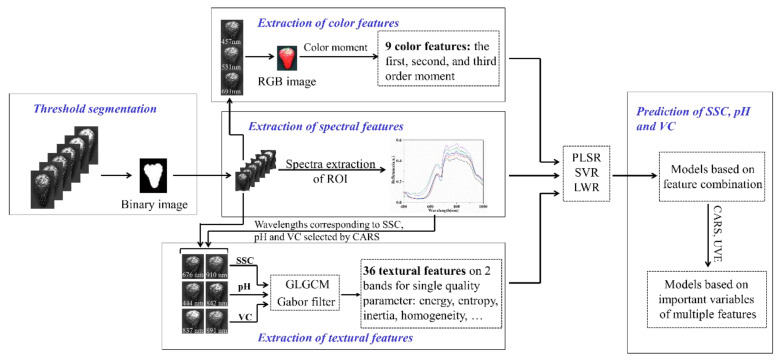
Flow diagram of prediction of soluble solid content (SSC), pH, and vitamin C (VC) of strawberries with multiple features. Partial least-squares regression (PLSR), support vector regression (SVR), locally weighted regression (LWR), gray level-gradient co-occurrence matrix (GLGCM), region of interest (ROI), competitive adaptive reweighted sampling (CARS) and uninformative variable elimination (UVE).

**Figure 2 sensors-20-03074-f002:**
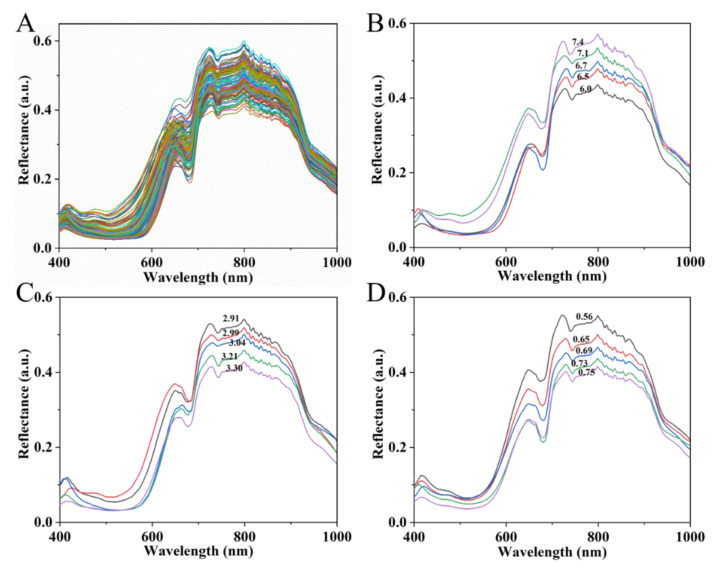
Reflectance spectra of strawberries. Spectra of 120 strawberries (**A**) and spectra of strawberries of different SSCs (**B**), pH (**C**), and VC (**D**) contents.

**Figure 3 sensors-20-03074-f003:**
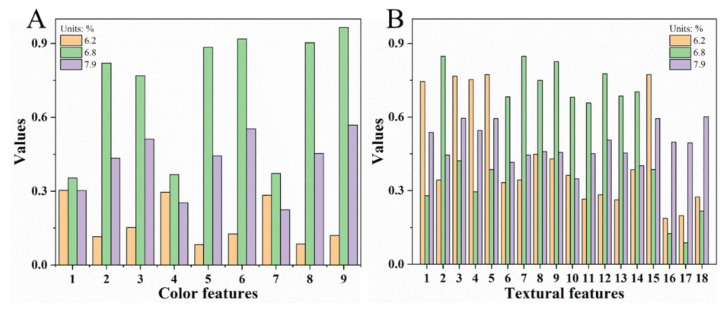
Color (**A**) and textural (**B**) features of strawberries with different SSCs.

**Figure 4 sensors-20-03074-f004:**
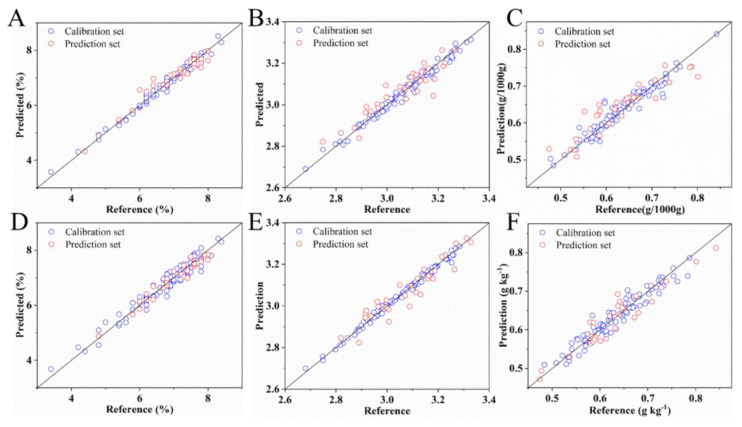
Prediction results of SSC (**A**), pH (**B**), and VC (**C**) using spectral, color, and textural features, and prediction results of SSC (**D**), pH (**E**), and VC (**F**) using important variables of multiple features.

**Figure 5 sensors-20-03074-f005:**
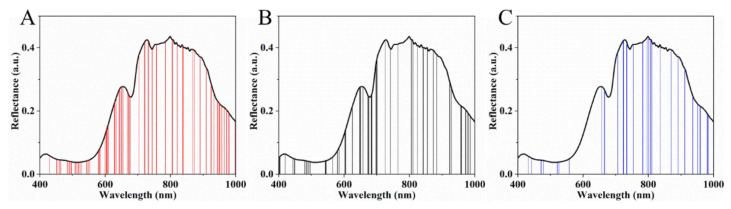
Important wavelengths selected by CARS for (**A**) SSC, (**B**) pH, and (**C**) VC.

**Figure 6 sensors-20-03074-f006:**
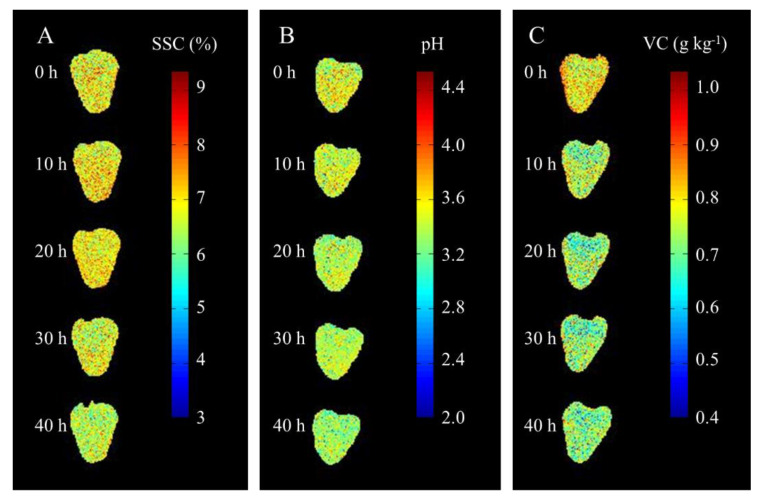
Distribution maps of SSC (**A**), pH (**B**), and VC (**C**) of strawberries during storage.

**Table 1 sensors-20-03074-t001:** SSC, pH, and VC statistics of 120 strawberries.

Parameters	Calibration			Prediction		
	Mean	SD	Range	Mean	SD	Range
SSC (%)	6.7734	0.8642	3.4–8.4	6.9668	0.9126	4.4–8.0
pH	3.0605	0.1347	2.68–3.33	3.0585	0.1290	2.75–3.26
VC (g kg^−1^)	0.6383	0.0706	0.4745–0.8421	0.6265	0.0673	0.4780–0.7832

**Table 2 sensors-20-03074-t002:** Prediction results of the SSC, pH, and VC of 120 strawberries by using spectroscopy. RMSEC = root mean square error of cross-validation; RMSEP = root mean square error of prediction; *R_c_*^2^ = determination coefficients of calibration; *R_p_*^2^ = determination coefficient of prediction.

Parameters	Methods	Pretreatment	*R_c_* ^2^	RMSEC	*R_p_* ^2^	RMSEP
SSC	PLSR	Raw	0.9769	0.0657	0.9044	0.1411
		WT	0.9776	0.0647	0.9182	0.1305
		MSC	0.9530	0.0971	0.8656	0.1465
	SVR	Raw	0.8772	0.1630	0.7867	0.2279
		WT	0.8600	0.1763	0.7748	0.2346
		MSC	0.8059	0.2171	0.7666	0.2426
	LWR	Raw	0.9999	0.0040	0.8850	0.1548
		WT	0.9999	0.0007	0.9364	0.1151
		MSC	0.9999	0.0104	0.8325	0.1635
pH	PLSR	Raw	0.9429	0.0322	0.8230	0.0543
		WT	0.9488	0.0305	0.8048	0.0570
		MSC	0.9131	0.0405	0.7162	0.0617
	SVR	Raw	0.9497	0.0303	0.8060	0.0588
		WT	0.9286	0.0362	0.7771	0.0669
		MSC	0.7435	0.0698	0.6598	0.0680
	LWR	Raw	0.9819	0.0181	0.8493	0.0501
		WT	0.9978	0.0063	0.8399	0.0516
		MSC	0.8530	0.0527	0.8237	0.0587
VC	PLSR	Raw	0.9698	0.0123	0.8216	0.0284
		WT	0.9587	0.0143	0.8245	0.0282
		MSC	0.9221	0.0182	0.8007	0.0318
	SVR	Raw	0.9628	0.0139	0.8035	0.0301
		WT	0.9494	0.0162	0.8204	0.0286
		MSC	0.9243	0.0180	0.8415	0.0247
	LWR	Raw	0.9912	0.0066	0.7841	0.0313
		WT	0.9998	0.0011	0.8144	0.0290
		MSC	0.9504	0.0145	0.8091	0.0295

**Table 3 sensors-20-03074-t003:** Prediction results of SSC, pH, and VC of 120 strawberries using spectral, color, and textural features.

Parameters	Methods	Features	*R_c_* ^2^	RMSEC	*R_p_* ^2^	RMSEP
SSC	PLSR	Spectroscopy + color	0.9800	0.0611	0.9370	0.1145
		Spectroscopy + texture	0.9951	0.0259	0.9299	0.1599
		Spectroscopy + color + texture	0.9984	0.0146	0.8746	0.2139
	SVR	Spectroscopy + color	0.9351	0.1104	0.8968	0.1498
		Spectroscopy + texture	0.8862	0.1612	0.7866	0.2282
		Spectroscopy + color + texture	0.9226	0.1243	0.8987	0.1541
	LWR	Spectroscopy + color	0.9998	0.0057	0.9294	0.1212
		Spectroscopy + texture	0.9999	0.0009	0.9172	0.1313
		Spectroscopy + color + texture	0.9993	0.0112	0.9106	0.1365
pH	PLSR	Spectroscopy + color	0.9618	0.0263	0.8230	0.0543
		Spectroscopy + texture	0.9822	0.0181	0.7110	0.0674
		Spectroscopy + color + texture	0.9831	0.0176	0.7087	0.0677
	SVR	Spectroscopy + color	0.9385	0.0341	0.8281	0.0562
		Spectroscopy + texture	0.9392	0.0333	0.7879	0.0638
		Spectroscopy + color + texture	0.9307	0.0355	0.7879	0.0632
	LWR	Spectroscopy + color	0.9682	0.0236	0.8404	0.0515
		Spectroscopy + texture	0.9701	0.0233	0.8343	0.0525
		Spectroscopy + color + texture	0.9687	0.0239	0.8401	0.0516
VC	PLSR	Spectroscopy + color	0.9685	0.0125	0.8039	0.0298
		Spectroscopy + texture	0.9567	0.0138	0.8769	0.0279
		Spectroscopy + color + texture	0.9515	0.0147	0.8423	0.0315
	SVR	Spectroscopy + color	0.9533	0.0155	0.7843	0.0312
		Spectroscopy + texture	0.9285	0.0193	0.8253	0.0275
		Spectroscopy + color + texture	0.9070	0.0217	0.7718	0.0322
	LWR	Spectroscopy + color	0.9907	0.0068	0.7651	0.0326
		Spectroscopy + texture	0.9896	0.0071	0.8444	0.0271
		Spectroscopy + color + texture	0.9950	0.0050	0.7708	0.0322

**Table 4 sensors-20-03074-t004:** Prediction results of SSC, pH, and VC of 120 strawberries using important variables of multiple features. UVE = uninformative variable elimination.

Variable Selection	Parameters	Methods	Features	Variables	*R_c_* ^2^	RMSEC	*R_p_* ^2^	RMSEP
CARS	SSC	PLSR	Spectroscopy + color	59 + 3	0.9445	0.1076	0.9431	0.0895
	pH	LWR	Spectroscopy	45	0.9934	0.0447	0.8858	0.0108
	VC	PLSR	Spectroscopy + texture	27 + 23	0.9228	0.0185	0.9109	0.0237
UVE	SSC	PLSR	Spectroscopy + color	46 + 3	0.9541	0.0959	0.8716	0.1432
	pH	LWR	Spectroscopy	57	0.8783	0.0554	0.7859	0.0474
	VC	PLSR	Spectroscopy + texture	31 + 28	0.9591	0.0134	0.9102	0.0251

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
