# Peer review of "Non-Destructive Detection of Strawberry Quality Using Multi-Features of Hyperspectral Imaging and Multivariate Methods"

_sensors, 2020, doi:10.3390/s20113074_

Round 1
Reviewer 1 Report
|
Main findings of the study. |
|
The main findings of the study concern the possibility, by mean of Hyperspectral imaging (HSI), to combine spectra, color and textural features for the analysis of Soluble solid content (SSC), pH, and vitamin C (VC) within strawberries. |
|
Limitations and strengths |
|
The strengths of the work lie just in the ability to construct the distribution maps of SSC, pH, and VC over storage time just by analyzing HIS spectra with the established regression model in a non-destructive way.
Limitations concern: 1) The English and writing level (see below); 2) The abstract seems too long and poorly focused. Here the authors should include also the mean values (as reported in Table 1) together with the goodness of the statistical analysis.
|
|
Methods, results and data interpretation |
|
The methods include essentially the use of HIS and statistical methods. Briefly, the average spectra were extracted from strawberries HIS and then the color and textural features were obtained using the color moment, the gray level-gradient co-occurrence matrix (GLGCM), and the Gabor filter. Finally, three regression algorithms were implemented to develop the regression model. So that, the authors were able to construct the distribution maps of SSC, pH, and VC over storage time just by analyzing spectra with the established model. The experimental procedures are well described, the results are sound and the data interpretation is convincing.
|
|
Detailed review report |
|
The authors developed a simple and non-destructive method for detecting Soluble solid content (SSC), pH, and vitamin C (VC) within strawberries in order to certificate their qualities. They use HIS features combined with regression models in order to construct the distribution maps of SSC, pH, and VC over storage time. This allowed the monitoring of strawberries properties in a non-destructive way. The results are sound but the work is not well written. In my opinion, as written above, the English level is too scarce so that also the scientific writing loses meanings and the paper is difficult to read. Therefore, corresponding revisions should be performed before the paper can be accepted on Sensors.
|
Author Response
Thanks for your review. The revised parts in out manuscript have labelled in red.
Question 1: The English and writing level (see below).
Answer: Thanks, it is a crucial question. We have modified some of the work in the manuscript.
“Then, PLSR, SVR, and LWR were used to develop the regression models to predict the SSC, pH, and VC of the strawberries.”
“For the prediction of SSC, PLSR with raw spectra revealed the prediction results: Rc2 = 0.9769, RMSEC = 0.0657, Rp2 = 0.9044, and RMSEP = 0.1411.”
“Textural features from the images of the first two largest weight wavelengths (Figure 1) selected by CARS (detailed parameter settings are shown in Table S5) were then extracted through the Gabor filter and GLGCM, and the selected wavelengths have the higher correlation with SSC, pH and VC. The textural features include mean, contrast and entropy, small grads dominance, big grads dominance, gray asymmetry, grads asymmetry, energy, gray mean, grads mean, gray variance, grads variance, correlation, gray entropy, grads entropy, entropy, inertia, and homogeneity; here, grads dominance can reflect the degree of grayscale change of the image.”
“Spectral, color, and textural features were integrated to develop the regression models established by PLSR, SVR and LWR for analysis of SSC, pH and VC.”
“With spectral and color features, PLSR obtains the optimal prediction of SSC with Rc2 = 0.9800, RMSEC = 0.0611, Rp2 = 0.9370, and RMSEP = 0.1145. However, the performance of LWR model declines with increasing features.”
“The figures indicate that the prediction errors of the three quality parameters are all low, and most of the data points are concentrated near the fitting line.”
“Selection of important variables from the above three features helps avoid these negative effects.”
“Specifically, the variables of the higher weight in spectral, color, and textural features were sought out.”
“The model developed by spectra of the 59 wavelengths selected by CARS (Figure 5A) and 3 color features of the first moment presented the best analysis results of Rc2 = 0.9445, RMSEC = 0.1076, Rp2 = 0.9431, and RMSEP = 0.0895 for SSC by PLSR. The optimal performance for pH was obtained from the LWR models developed by spectra of the 45 wavelengths selected by CARS (Figure 5B) with Rc2 = 0.9934, RMSEC = 0.0447, Rp2 = 0.8858, and RMSEP = 0.0108.”
“For VC, the lowest prediction errors are Rc2 = 0.9228, RMSEC = 0.0185, Rp2 = 0.9109, RMSEP = 0.0237, and the results are got from PLSR model developed by the spectra of 27 wavelengths selected by CARS (Figure 5C) and 23 textural features. The 23 features include 18 textural features from the image of 837 nm and low grads dominance, grads asymmetry, gray variance, correlation, and grads entropy from the image of 891 nm.”
“The figures show that the prediction errors of the optimal models of SSC, pH and VC were smaller after variable selection.”
Question 2: The abstract seems too long and poorly focused. Here the authors should include also the mean values (as reported in Table 1) together with the goodness of the statistical analysis.
Answer: Thanks, your advice is very pertinent. We have revised the abstract and added the depiction of statistical analysis.
“Thus, the proposed method can nondestructively and accurately determine SSC, pH, and VC of strawberries and is expected to design and construct the simple sensors for the above quality parameters of strawberries.”
“SSC, pH and VC of 120 strawberries were statistically analyzed to facilitate the partitioning of data sets, which helped optimize the model.”
Reviewer 2 Report
In this paper, the spectral, color, and textural features extracted from hyperspectral images were investigated for the non-destructive detection of SSC, pH, and VC in strawberries. PLSR, SVR, and LWR were adopted to develop regression models for determining SSC, pH, and VC using these three types of features, and WT was used to reduce the noise of the spectra. After careful evaluation of the paper, the reviewer believes this paper provides novelty, contribution and analysis. However, I have several suggestions and concerns:
(1) This paper only uses several linear regression methods to predict SSC, pH, and VC. I would suggest the authors can also apply other non-linear methods such as kernel support vector machines and deep learning algorithms.
(2) In the data pre-processing, I would suggest apply Multiplicative scatter correction (MSC) to correct spectra in such a way that they are as close as possible to a reference spectrum.
(3) Band selection is the important issue in the HSI, I suggest the authors can compare other BS methods with the extraction of color features.
(4) In the tables of prediction results, the authors can highlight the best methods in the table to make readers can easy understand the performance.
Author Response
Thanks for your review. The revised parts in out manuscript have labelled in red.
Question 1: This paper only uses several linear regression methods to predict SSC, pH, and VC. I would suggest the authors can also apply other non-linear methods such as kernel support vector machines and deep learning algorithms.
Answer: Thanks for your advice. We have tried some nonlinear algorithms in the data processing process, such as random forest (RF) and convolutional neural network (CNN), but the effect was not good. And linear algorithms are helpful for features optimization and analysis. Thus, the nonlinear method was not used in the following study.
Question 2: In the data pre-processing, I would suggest apply Multiplicative scatter correction (MSC) to correct spectra in such a way that they are as close as possible to a reference spectrum.
Answer: Thanks, your advice is to the point. We have used MSC to pretreat spectra and supplemented the results with the Table 2. The analysis of the results has been added in the manuscript.
Question 3: Band selection is the important issue in the HSI, I suggest the authors can compare other BS methods with the extraction of color features.
Answer: Thanks, your advice is very pertinent. We have used uninformative variable elimination (UVE) to select the important variable. And CARS and UVE were compared. The corresponding results are shown in Table 4. The analysis of the results has been added in the manuscript.
Question 4: In the tables of prediction results, the authors can highlight the best methods in the table to make readers can easy understand the performance.
Answer: Thanks for your advice. We have shown the best results in bold.
Reviewer 3 Report
Hyperspectral Imaging (HSI) is a sensing technique that captures and visualizes the reflected light of an object with high resolution at a number of wavelengths ranging from tens to hundreds. In this study, the multiple absorption points at wavelengths correlated with SSC, pH, and VC were found from HSI measurements, and visualized images of these distributions and graphs of the distribution ratios were prepared, and inter-individual evaluation was attempted to confirm the usefulness of this method. The progressiveness of this study is that the accuracy was enhanced by using both spectral and textural information. The prediction accuracy evaluated from the determination coefficients of prediction are good, and it is expected to be able to provide practical sensing technology. The experiment was successfully performed and the data were well obtained. I can recommend that this paper be published as it is.
Author Response
Thanks for your review.
Comments and Suggestions for Authors:
Hyperspectral Imaging (HSI) is a sensing technique that captures and visualizes the reflected light of an object with high resolution at a number of wavelengths ranging from tens to hundreds. In this study, the multiple absorption points at wavelengths correlated with SSC, pH, and VC were found from HSI measurements, and visualized images of these distributions and graphs of the distribution ratios were prepared, and inter-individual evaluation was attempted to confirm the usefulness of this method. The progressiveness of this study is that the accuracy was enhanced by using both spectral and textural information. The prediction accuracy evaluated from the determination coefficients of prediction are good, and it is expected to be able to provide practical sensing technology. The experiment was successfully performed and the data were well obtained. I can recommend that this paper be published as it is.
Answer: Thank you for your recognition of our research work.
Round 2
Reviewer 2 Report
After reviewing the revised version, I accept in present form.